# Undergraduates' lifestyle and Suboptimal Health Status (SHS): A cross-sectional study in the Ha'il region of Saudi Arabia

Bandar Alsaif[1], Collins Otieno Asweto[2]°, Sehar-un-Nisa Hassan[1]°*, Mohamed Ali Alzain[1,3]°, Mohammed Elshiekh Saeed[4,5], Ahmed Kassar[1], Kamal Elbssir Mohammed Ali[6], Mouna Ghorbel[7], Rafat Zrieq[1,8], Wei Wang[9,10]

**1** Department of Public Health, College of Public Health and Health Informatics, University of Ha'il, Ha'il, Saudi Arabia, **2** Department of Community Health, School of Nursing, University of Embu, Embu, Kenya, **3** Department of Community Medicine, Faculty of Medicine and Health Sciences, University of Dongola, Dongola, Sudan, **4** Faculty of Medicine, National University-Sudan, Khartoum, Sudan, **5** Department of Physiology, Faculty of Medicine, University of Dongola, Dongola, Sudan, **6** Department of Community Health, Occupation Health and Safety Program, Northern Border University, Arar, Saudi Arabia, **7** Department of Biology, College of Sciences, University of Hail, Ha'il City, Saudi Arabia, **8** Applied Science Research Centre, Applied Science Private University, Amman, Jordan, **9** Clinical Research Center, First Affiliated Hospital, Shantou University Medical College, Shantou, China, **10** Centre for Precision Health, Edith Cowan University, Joondalup, Western Australia, Australia

° These authors contributed equally to this work.
* s.nisa@uoh.edu.sa

## Abstract

### Background

University students in Saudi Arabia are embracing some of the negative traits of the fast-paced modern lifestyle, typified by unhealthy eating, low physical activity, and poor sleep habits that may increase their risk for poor health. Health and holistic well-being at the population level are among the priorities of the 2030 vision of a vibrant society in the Kingdom of Saudi Arabia. The current study thus aims at determining the prevalence and predictive factors of Suboptimal Health Status (SHS) among university students.

### Methods

A cross-sectional study was conducted among 9,026 undergraduate students between 31st May and 15th June 2023. The data was collected through an online questionnaire using the Arabic translation of the SHS (ASHSQ-25) and the Simple Lifestyle Indicator Questionnaire (SLIQ). The chi-square test, Canonical Correlation Analysis (CCA), hierarchical multiple regression, and Receiver Operating Characteristic (ROC) were performed to analyze the data using IBM SPSS software (version 25.0) at a significance level of p<0.05.

### Results

The findings showed that 33.7% (3038) of the students were classified as having SHS. The prevalence of SHS was statistically higher among younger and those studying social science discipline, overweight, had poor diets, engaged in low physical activity, and slept <6

**Data Availability Statement:** Data Availability Statement: All relevant data are within the manuscript and its Supporting Information files.

Access to anonymous data used for this analysis is submitted along with this article and available at https://www.kaggle.com/datasets/drseharunnisahassan/lifestyle-and-suboptimal-health/data. Further information about data can be obtained upon direct request sent via e-mail to either the corresponding author on this manuscript or the secretary of training at the College of Public Health and Health Informatics at: training.cphhi@uoh.edu.sa.

**Funding:** This research was funded by the Deanship of Scientific Research at the University of Ha'il, funding number RG-20 215. The funders had no role in study design, data collection and analysis, decision to publish, or preparation of the manuscript.

**Competing interests:** The authors have declared that no competing interests exist.

hours (p<0.001). Canonical loadings of these factors were significant in predicting the five SHS dimensions range (0.52–0.97). Furthermore, the extracted significant variables from the multiple regression analysis indicated the final model (Model 3) was statistically significant, $R^2 = 0.646$, $F (6,9019)$, $p <0.0001$, suggesting that 64.6% of the variance in the SHS can be explained by the predictor variables in the model. These variables included age, college, academic level, diet, BMI, and sleeping duration.

## Conclusion

Findings show that modifiable predictors such as poor sleep quality, higher BMI scores, and poor dietary habits increase SHS risk among university students in Saudi Arabia. Therefore, the findings of this study emphasize the necessity of early interventions that promote healthy lifestyles.

## 1. Introduction

The demographic profile of the population in Saudi Arabia reveals that youth represents more than one-third of the Saudi population. Saudi Arabia has great expectations from this large segment of population to make significant contributions to the social and economic development of the country to successfully achieve the targets set for Vision 20230. Therefore, Saudi Arabia is making significant investments in youth education and employment programs like other developed economies [1]. Higher academic achievement has opened doors for advanced career development and better social status for youth, nonetheless, university students are exposed to various study-related stressors and social pressures that may impact their health and wellness. Investing in youth health is just as important as investing in their education and employment. Technology has a significant impact on the lifestyle and ecological environment of youth. The rapid evolution and deployment of digital tools in higher education have been beneficial to achieving learning outcomes and have increased the independence of students in accomplishing their higher educational goals. However, these transitions have presented other challenges such as a distracted mind, difficulty in focus and fatigue that are attributed to less sleep time and increased screen time [2]. Lifestyle factors have been found to impact health and wellness in youth [3]. Public health experts and organizations emphasize accumulating evidence to understand the complex dynamics at play in shaping public health outcomes related to suboptimal health status. This insight will be useful to overcome the long-term impact of these problems on the vitality, educational progress and overall health of students. Findings will be useful for higher education institutes in developing effective preventive interventions for promoting youth health.

### 1.1 Theoretical framework on population health and SHS

Suboptimal health status has its basis in Traditional Chinese Medicine (TCM) where the human body's health conditions do not meet the diagnostic criterion for any disease, however, it is also not working in its optimal conditions. It is positioned somewhere between well-being and illness preceding the occurrence of chronic diseases. The state of SHS has been illustrated in literature as a range of physical and mental symptoms experienced by a person that may persist for at least three months and cause generalized fatigue and reduced immunity to fight against diseases thus act a precursor of chronic health conditions such as diabetes, cardiovascular disease, and mental health disorders [4]. Lifestyle and environmental factors influence

population health due to their association with sub-optimal health which then significantly increases the risk for non-communicable diseases (NCDs) [5]. The field of public health has realized the heightened relevance of suboptimal health status (SHS) in disease prevention and health promotion efforts in recent times. It is a crucial indicator for population health as it highlights the need for early intervention and personalized medical care to prevent the progression to more serious health issues. By addressing suboptimal health status in its early stages, overall health outcomes can be improved, and the burden of chronic disease on society can be reduced. Current research suggests that understanding and addressing suboptimal health status can lead to more proactive and effective population health management strategies, ultimately improving the well-being of the community as a whole [6, 7].

Suboptimal Health Status Questionnaire (SHSQ-25) is a self-report measure that has been well-tested for its psychometric properties and used in the assessment of SHS [8–11]. The tool was initially constructed by Yan et al. in 2009 [12] and has been translated into various languages. It encompasses 25 questions distributed across five domains: mental health, immune system, fatigue, digestive tract and cardiovascular health. It has been validated among major three regions, namely in Asian [12], European [13], African [14] and most recently, Arab population [15].

## 1.2 Insights and gaps in existing literature

In the past few years, research studies from some regions of the world [5, 6] but mainly from China have focused on exploring SHS in general populations [7, 8] and more specifically on university students [9, 10]. Most of the available studies that explored predictors of SHS among student populations were carried out in Chinese universities and reported that rates of SHS were around 56% and unhealthy lifestyles were significant determinants of SHS [16]. A large-scale community-based study collected data from 1461 students studying in universities located in three discrete zones of China and found that over fifty per cent of students experienced SHS. More specifically, one-quarter of students reported physical symptoms of SHS, and the rates of psychological symptoms were twice as high. Factors such as inadequate sleep time or poor quality of sleep, study burden, an insufficient amount of physical activity, alcohol consumption, and smoking were significant risk factors for SHS while seeking help, engagement in social activities, and daily one-hour of internet access were identified as protective factors. Findings from this research recommended that adolescent populations be directed to adopt and integrate health activities into daily lives to prevent risks for short and long-term repercussions on health [17]. Another research on Chinese medical students [18] found that 42.8% experienced SHS, which was elevated than the degree of SHS reported in community samples. This study found that short and poor sleep quality were independently and jointly associated with SHS. In contrast, better sleep quality was associated with lower proportions of SHS [18]. The key weakness of this research was that it did not assess the long-term effects of sleep duration on SHS and primarily relied on sleep patterns in the past five consecutive days.

A comparative investigation of differences in the lifestyle factors among international and Chinese students and the impact of these factors on their health status demonstrated that Chinese students had relatively better scores on indicators of healthy lifestyle. Additionally, three aspects of the lifestyle, namely exercise, nutritious diet, and meaningful social connections, were found to be significantly related to health status among university students. Participants with a moderate Health-Promoting Lifestyle Profile (HPLP) were substantially at higher risk of experiencing negative outcomes on their health as compared to those with an excellent HPLP [19]. Some studies from Saudi Arabia have investigated the lifestyle factors that may impact health status among university students and found that excessive speeding and lack of physical exercise were the most prevalent health-risk behaviours [20, 21]. Despite these studies

recommending the preparation and execution of programs to encourage university students to adopt a proactive approach to adopting healthy lifestyles, there is a slow response from all stakeholders at the governmental and institutional levels, and students need to be more engaged in healthy lifestyles [22].

Saudi Arabia is keen to develop and expand its higher educational resources and encourage international students to join higher educational institutes in Saudi Arabia that in the long term will be useful for the economic and social development of the country. There is increased emphasis on the development and improvement of the curriculums, with an emphasis on outcome-based learning and incorporating emerging realms in the use of technology in higher education [23]. There is also a requirement for research-informed policies and programs that should improve lifestyle factors responsible for poor health among university students [22, 24]. Addressing the current gaps in research concerning SHS and the related factors affecting university students in Saudi Arabia is thus warranted.

Therefore, in this quantitative research, we tried to answer the main question: What are the predictive factors that impact SHS experienced by students studying in higher education in Saudi Arabia? Identifying predictors will help close gaps in research in this area. Besides, these insights will help develop measures to improve overall community health outcomes and prevent diseases, aligning with Saudi Arabia's Vision 2030 goals.

## 2. Methods

### 2.1 Study design, target population and study sample

The present study was completed employing a quantitative cross-sectional research design. The data was collected during the timeframe of 31$^{st}$ May and 15$^{th}$ June 2023, by surveying university students with minimum age of 18 years. The data was collected using an online questionnaire.

The target population included students enrolled in bachelor's programs at higher education institutions in Ha'il City, Saudi Arabia. The least estimated size for choosing a sample for the current study was 6947, which was determined by utilizing the given equation [25]:

$$\mathbf{n} = \frac{Z^2 P (1-P)}{d^2}$$

Where n is the sample size, Z is 1.96 (the number equivalent to the 95%CI), P is the expected prevalence of 23.7% [15], and d is precision equal to 0.01.

The questionnaire link was distributed to the University students via various online portals (WhatsApp, Twitter, and email). The number of responses in total was 10,775, with a response rate of 90.5%. Subsequently, 1,749 respondents were excluded because they had chronic diseases, psychological problems, or missing some data. Furthermore, a total of (n = 9,026) eligible students were included for further analysis.

### 2.2 Ethical considerations

The present study considered the ethical principles following the Declaration of Helsinki. The information about the study's objectives, voluntary participation, and freedom to withdraw was shared with respondents via electronic informed consent. Furthermore, the confidentiality and anonymity of participants' answers were ensured during data collection and reporting. The study was conducted in accordance with the Declaration of Helsinki, and approved by the Institutional Ethical Review Board at the University of Ha'il (approval letter number H-2021-221 dated 6 Nov, 2021)".

### 2.3 Data collection

The study questionnaire contained the following tools:

**2.3.1 Arabic version of SHS (ASHSQ-25).** Recently, our team translated the SHSQ-25 scale into Arabic and validated it on the Saudi population; demonstrating that the tool is reliable and culturally appropriate for assessing SHS in the general population [15]. The ASHSQ-25 has adequate psychometric properties with (Cronbach's α = 0.91) and item internal consistency (IIC) on the four domains of SHSQ 25 (Fatigue, Cardiovascular, Mental health and Digestive) located between 0.49 to 0.96, underscores the scale's consistency and dependability [15]. The cut-off point for determining SHS was a score >33, which represents "SHS", while a score of ≤33 indicates " healthy" [15].

**2.3.2 Simple Lifestyle Indicator Questionnaire (SLIQ).** This scale was used to measure various components of lifestyle that may affect the health status of the students. The psychometric properties of SLIQ have been confirmed in different research studies [26, 27]. The scale comprises 12 questions across five domains: firstly, the diet domain includes three questions, each with a score range from (0–5); A sum diet category scores <5 is considered poor diet, while a range (6–10) is considered moderate diet, and >10 is considered a good diet. Secondly, the physical activity domain includes three questions on light, moderate, and vigorous exercise. Respondents are classified as having light exercise if they only engage in light exercise, moderate exercise if they engage in any moderate activity, and vigorous exercise if they engage in any vigorous activity. Thirdly, the alcohol consumption domain includes three questions. Finally, the smoking domain includes two questions, and there is one additional question related to life stress [26, 27].

**2.3.3 Factors affecting SHS.** The survey questionnaire includes demographic and academic factors. These factors included gender, age, academic level, study discipline, Body Mass Index (BMI) and sleep duration.

### 2.4 Data analysis

The statistical software IBM SPSS, version 25.0, was utilized for storing and analyzing data. Descriptive statistics, such as percentage values, were computed to describe the categorical variables. The chi-square test was utilized to examine the association between demographic and lifestyle factors with SHS, by choosing p-value significance at p<0.05. Canonical Correlation Analysis (CCA) was done to identify relationships between latent variables, such as students' lifestyles, characteristics, and SHS within variable sets. Since CCA can merge the observed dependent and independent variables to form a single synthetic (or latent) construct, it offers the highest possible simple correlation between two groups of study variables [28].

Furthermore, a hierarchical multiple regression analysis determined the unique contribution of each set of demographic and lifestyle factors in SHS [29]. These interpretations aid in comprehending the hierarchical structure of predictor variables and their impact on health status. Receiver Operating Characteristic (ROC) was executed to determine the accuracy of the predictive ability of demographic and lifestyle factors.

## 3. Results

### 3.1 SHS of participants across demographic and lifestyle variables

The study comprised 9,026 university students, 52.2% of whom were males, with 33.7% (3038) classified as having SHS. The prevalence of SHS was statistically significantly different between genders, with 41.5% (1791) of females and 26.4% (1247) of males. In addition, SHS was significantly increased with decreased age; students aged 18–20 years were at a higher risk of experiencing SHS at 47.3%, followed by those aged between 21–25 and >25 years at 37.2% and

**Table 1. Distribution of Suboptimal Health Status (SHS) across demographics and lifestyle factors (n = 9,026).**

| Variables | Categories | Total n (%) | Healthy n (%) | SHS(a) n (%) | Value of the test | p value |
|---|---|---|---|---|---|---|
| **Overall** | | 9026 | 5988(66.3) | 3038(33.7) | - | - |
| **Age groups** | 18–20 years | 1225(13.6) | 645(52.7) | 580(47.3) | 737.86 | <0.001 |
| | 21–25 years | 6593(73.0) | 4140(62.8) | 2453(37.2) | | |
| | >25 years | 1208(13.4) | 1203(99.6) | 5(0.4) | | |
| **Gender** | Male | 4715(52.2) | 3468(73.6) | 1247(26.4) | 229.87 | <0.001 |
| | Female | 4311(47.8) | 2520(58.5) | 1791(41.5) | | |
| **Academic Level** | First | 659(7.3) | 452(68.6) | 207(31.4) | 1877.1 | <0.001 |
| | Second | 1619(17.9) | 843(52.1) | 776(47.9) | | |
| | Third | 1847(20.5) | 631(34.2) | 1216(65.8) | | |
| | Fourth | 1911(21.2) | 1665(87.1) | 246(12.9) | | |
| | Fifth | 2008(22.2) | 1800(89.6) | 208(10.4) | | |
| | Internship | 982(10.9) | 597(60.8) | 385(39.2) | | |
| **College** | Health | 2688(29.8) | 2251(83.7) | 437(16.3) | 1065.3 | <0.001 |
| | Engineering | 3584(39.7) | 2549(71.1) | 1035(28.9) | | |
| | Social Sciences | 2754(30.5) | 1188(43.1) | 1566(56.9) | | |
| **BMI** | Underweight | 16(0.2) | 6(37.5) | 10(62.5) | 2431.2 | <0.001 |
| | Normal | 6059(67.1) | 5058(83.5) | 1001(16.5) | | |
| | Overweight | 2928(32.4) | 911(31.1) | 2017(68.9) | | |
| | Obese | 23(0.3) | 13(56.5) | 10(43.5) | | |
| **Diet Category** | Poor diet | 5445(60.3) | 2626(48.2) | 2819(51.8) | 2019.2 | <0.001 |
| | Moderate diet | 3376(37.4) | 3159(93.6) | 217(6.4) | | |
| | Good diet | 205(2.3) | 203(99.0) | 2(0.1) | | |
| **Activity category** | Light | 3089(34.2) | 1462(47.3) | 1627(53.6) | 813.2 | <0.001 |
| | Moderate | 575(6.4) | 360(62.6) | 215(37.4) | | |
| | Vigorous | 5362(59.4) | 4167(77.7) | 1196(22.3) | | |
| **Sleeping hours** | Less than 6 hours | 1291(14.3) | 239(18.5) | 1052(81.5) | 1908.5 | <0.001 |
| | 6 hours | 2221(24.6) | 2008(90.4) | 213(9.6) | | |
| | 6–9 hours | 5498(60.9) | 3734(67.9) | 1764(32.1) | | |
| | >9 hours | 16(0.2) | 7(43.8) | 9(56.2) | | |

4%, respectively. Besides, being in the third year 65.8% (1216), Social science students 56.8% (1565), overweight 68.9% (2016), poor diet 51.8% (2819), low physical activity 52.6% (1625), and sleeping <6 hours 81.5% (1052) were associated to the higher prevalence of SHS at p<0.001 as shown in "Table 1".

## 3.2 Relationships between lifestyle behaviors, BMI, and SHS

The whole CCA model showed relationships between predictor variables (lifestyle and BMI) and criterion variables (fatigue, immunological system, digestive system, circulatory system, and mental state). Five functions were generated in the CCA model [Wilks' $\lambda = 0.761$, $F_{(5, 3463)} = 44.67$, $p < 0.001$], indicating a substantial correlation between lifestyle behaviors and BMI with these five SHS features. The model's variance was explained by 31.2% with an effect size of 0.312 ($r^2$-type). $Root^2$ was excluded from the analysis because its $Rc^2$ value was less than 0.1, which did not meet Tabachnick and Fidell's (2014) cut-off criteria for the $Rc^2$ effect [30]. This decision was based on statistical results showing Wilks' $\lambda = 0.769$, $F_{(2, 1657)} = 25.00$, $p < 0.01$.

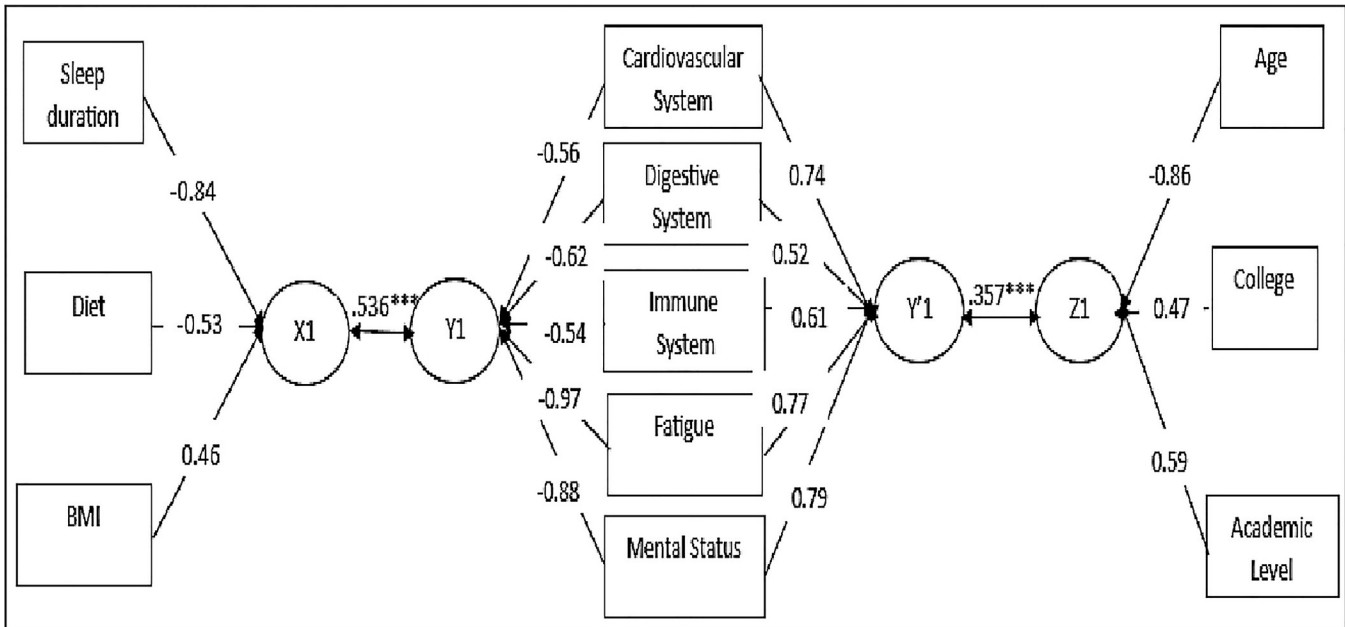

**Fig 1. The relationships between students' characteristics, lifestyle choices, and SHS domains.** Z1 is a canonical variable for student characteristics; X1 is a canonical variable for lifestyle behaviour and BMI; Y1, Y'1 are canonical variables for SHS domains. Present are only variables with canonical loadings greater than 0.4.

The canonical loadings show how each variable is connected to the canonical variable. The criterion of 0.4 was utilized to identify important variables in the two sets as indicated [31]. Sleep duration, BMI, and diet were the most important predictive variables. The five SHS dimensions—fatigue, immunological system, digestive system, circulatory system, and mental state—were essential criteria. Specifically, SHS had a negative correlation with sleep duration (rs = -0.84) and diet (rs = -0.53) but a positive correlation with BMI (rs = 0.46), as shown in "Fig 1" and "Table 2".

### 3.3 Relationships between student characteristics and SHS

Additional CCA was done to find possible connections between SHS and student's attributes. The only important function that came out of the results was Wilks' λ = 0.756, F (3, 2831) = 54.00, p<0.001. It explained 34.7% of the difference between the two factors. The linear combination of SHS factors and the combination of student characteristics showed a weak link (p < 0.001), as shown by the canonical correlation of Root 1, which was 0.357.

The canonical loadings for the five SHS parameters were over the 0.4 cut-off [31]. Only age, college, and academic level among student characteristics reached this threshold. The findings indicated a negative correlation between age (rs = -0.86) and positive correlations between respondents' academic level (rs = 0.59) and fatigue levels (rs = 0.77), as well as their cardiovascular system (rs = 0.74), digestive system (rs = 0.52), immune system (rs = 0.61), and mental state (rs = 0.79) as shown in "Table 3" and "Fig 1".

### 3.4 Predictors of SHS among the students

The ROC analysis found that the area under the curve (AUC) ranged from 64.3% to 83.5%, indicating good overall predictive accuracy and suggests that the lifestyle variables accurately predicted SHS, as shown in "Fig 2".

**Table 2. Association between lifestyle behaviors and Suboptimal Health Status (SHS).**

| Variables | Function 1 | | |
|---|---|---|---|
| | $r_{s\,(a)}$ | $r_s^2\,_{(b)}$ | $h^2\,_{(c)}(\%)$ |
| **Predictor variable–Life state** | | | |
| Daily physical activity | 0.34 | 11.84 | 12.64 |
| Sleep duration | -0.84 | 70.42 | 71.06 |
| Diet | -0.53 | 27.61 | 27.43 |
| Stress | 0.26 | 6.65 | 6.32 |
| BMI | 0.46 | 20.98 | 21.24 |
| **Criterion variables- SHS** | | | |
| Cardiovascular system | -0.56 | 31.42 | 31.53 |
| Digestive system | -0.62 | 38.56 | 39.02 |
| Fatigue | -0.97 | 93.68 | 92.87 |
| Mental status | -0.88 | 76.99 | 77.05 |
| Immune system | -0.54 | 29.34 | 30.21 |
| Canonical correlation | 0.531***(d) | | |

rs = structure coefficients (canonical loadings)

rs² = squared structure coefficients

h² = communality coefficient.

***$p < 0.001$.

"Table 4" presents the findings from hierarchical multiple regression analysis. Based on model 3 of hierarchical multiple regression analysis, age, college, academic level, diet, and BMI were predictors of SHS among students. The full model (Model 3) was statistically significant, $R^2 = 0.646$, $F\,(6, 9019) = 2743.165$, $p < .0001$, indicating that 64.6% of the variance in the SHS can be explained by the predictor variables in the model.

**Table 3. Correlation between students' characteristics and SHS.**

| Variables | Function 1 | | |
|---|---|---|---|
| | $r_{s\,(a)}$ | $r_s^2\,_{(b)}$ | $h^2\,_{(c)}(\%)$ |
| **Predictor variable–Student's characteristics** | | | |
| **Age** | **-0.86** | 74.24 | 74.46 |
| **Gender** | 0.21 | 4.45 | 4.53 |
| **Academic level** | **0.59** | 34.92 | 34.83 |
| **College** | **0.47** | 22.03 | 21.97 |
| **Criterion variables- SHS** | | | |
| **Cardiovascular system** | **0.74** | 54.66 | 54.58 |
| **Digestive system** | **0.52** | 27.14 | 27.09 |
| **Fatigue** | **0.77** | 59.39 | 60.23 |
| **Mental status** | **0.79** | 62.55 | 62.47 |
| **Immune system** | **0.61** | 37.17 | 37.25 |
| **Canonical correlation** | 0.357*** (d) | | |

rs = structure coefficients (canonical loadings)

rs² = squared structure coefficients

h² = communality coefficient.

***$p < 0.001$.

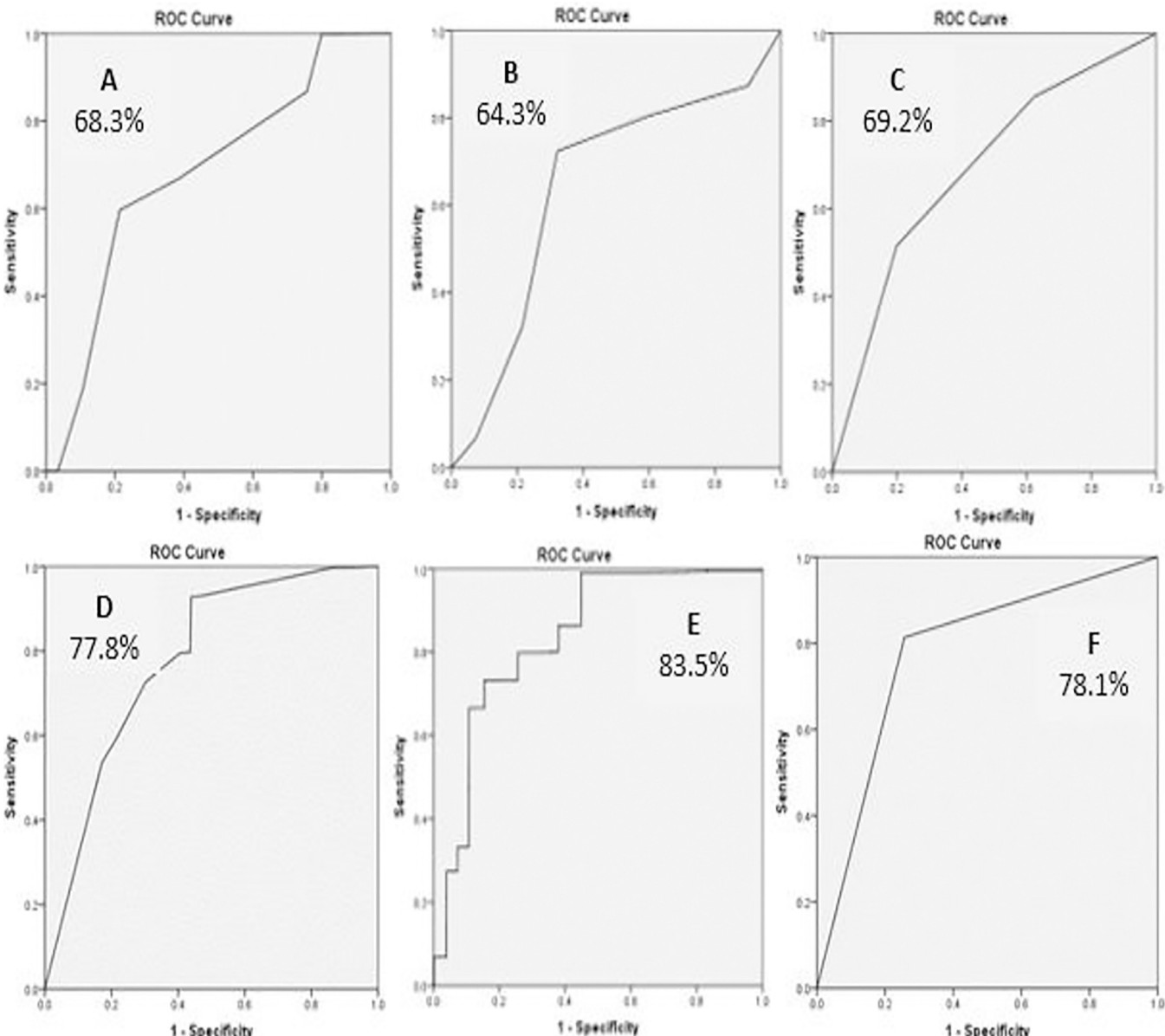

**Fig 2. ROC curve results for the six predictors of SHS.** A–Age (Area under curve = 68.3%), B–Academic level (Area under curve = 64.3%), C–College (Area under curve = 69.2%), D–Diet (Area under curve = 77.8%), E–BMI (Area under curve = 83.5%), F- Sleeping hours (Area under curve = 78.1%).

From the model, age (β = -0.435, p< 0.001) and sleeping duration (β = -0.468, p< 0.001) stood out to as the most impactful predictors of SHS, albeit negatively correlated to SHS as shown in "Table 4".

The regression equation: $Y = b_0 + b_1x_1 + b_2x_2 + b_3x_3 + b_4x_4 + b_5x_5 + b_6x_6 + e$ . . . . . . . . . . . ...(1)

Where, Y is SHS score, $b_1$ coefficient for age, $b_2$ for college, $b_3$ for academic level, $b_4$ for dietary, $b_5$ for BMI, and $b_6$ for Sleeping duration. While e is for residual term. Therefore, the equation will be:

SHS score = 84.414–3.973 (Age) + 3.813 (College) + 3.79 (Academic level) - 8.4 (Diet category) + 1.83 (BMI) -10.28 (Sleeping duration) . . . . . . . . . . . . . . . . . . . . . . . . . . . . . . . . . .. (2)

**Table 4. Prediction of SHS through hierarchical multiple regression (n = 9,026).**

| Variables | SHS score | | | | | |
|---|---|---|---|---|---|---|
| | Model 1 | | Model 2 | | Model 3 | |
| | B (a) | β (b) | B | β | B | β |
| Constant | 119.24*** | | 89.76*** | | 84.414*** | |
| Age | -3.91*** | -0.428*** | -3.68*** | -0.402*** | -3.973*** | -0.435*** |
| College | | | 8.413*** | 0.405*** | 3.813*** | 0.184*** |
| Academic level | | | 1.98*** | 0.179*** | 3.79*** | 0.342*** |
| Diet category | | | | | -8.4*** | -0.280*** |
| BMI | | | | | 1.83*** | 0.327*** |
| Sleeping duration | | | | | -10.28*** | -0.468*** |
| R | 0.428 | | 0.562 | | 0.804 | |
| R² | 0.183 | | 0.316 | | 0.646 | |
| ΔR² | 0.183 | | 0.133 | | 0.330 | |
| ΔF | 2025.2*** | | 874.82*** | | 2803.11*** | |

(a)unstandardized coefficient

(b) standardized coefficient

*** p<0.001

## 4. Discussion

This study found that one-third of the students experienced SHS, and it was conspicuously high among students who were young adults, females, studying social sciences disciplines, either underweight or overweight, had poor diet, performing light to moderate physical activity, and having either longer or shorter sleeping hours. The CCA findings revealed that lifestyle variables such as sleep duration, BMI, and poor diet were the most important predictive variables to the five SHS dimensions (fatigue, immunological system, digestive system, circulatory system, and mental state). Similarly, age, college, and academic level were also important predictive variables to the five SHS dimensions. For instance, SHS showed a negative relationship with both the amount of sleep and diet but a positive relationship with BMI. These findings suggest that a decrease in sleep duration and consumption of an unhealthy diet are linked to higher SHS scores, while an increase in BMI is linked to an increase in SHS scores. The overall findings from hierarchical multiple regression analysis indicate that 64.6% of the variance in the SHS among undergraduate students can be explained by the sixth predictor variables in the model.

Prior research by Ma et al. (2018) has established a correlation between optimal sleep quality, ample physical activity, sufficient nutrient intake, and reduced scores on the SHS scale [2]. Moreover, there is a correlation between inadequate sleep quality and decreased levels of physical activity, which is recognized as key component in the development of cardiovascular and cerebrovascular diseases [32]. Sleep is a naturally occurring and reversible condition characterized by decreased receptivity to external stimuli, relative inactivity, and a loss of consciousness. Insufficient sleep and disturbances in sleep patterns can lead to significant cognitive and emotional impairments [33, 34]. Prior research has indicated that insufficient sleep duration is associated with an increased risk of cardiovascular disease, cerebrovascular disease, mental disorders, and obesity [35–37]. These findings suggest that inadequate sleep quality is associated with an increased risk of SHS among students, which is consistent with our observation.

In terms of diet, Xu et al. (2020) discovered a strong correlation between poor dietary choices, inconsistent meal schedules, and an increased occurrence of SHS [38]. The

importance of diet and nutrition in promoting mental health and overall well-being was confirmed by the findings of Owen et al. in 2017 [39]. Furthermore, a strong correlation exists between a diet heavy in salt and hypertension [40]. Cordner and colleagues have presented evidence supporting the recognized connections between consuming a diet high in fat or commonly referred to as a "Western" diet and the development of conditions like obesity, diabetes, and cardiovascular disease [41]. Furthermore, a substantial amount of evidence indicates that diets rich in fat can significantly affect the brain, behavior, and cognition [41]. Moreover, inadequate dietary habits have been documented among university students in various countries [42], as well as several other affluent nations [43]. These findings corroborate our current findings indicating that the students have an inadequate diet, which puts them at high risk of suboptimal health.

Weight gain is common among students as they transition into university life. This era is essential because young adults can make independent eating choices, which can lead to changes in their dietary habits [44, 45]. Gan et al. (2011) have demonstrated that certain groups of individuals are more susceptible to adopting poor eating habits characterized by insufficient nutrient intake [46]. Several of these behaviors encompass inconsistent meal patterns, omission of breakfast, diminished consumption of fruits and vegetables, and heightened intake of fried food [47]. In addition to food changes, inadequate exercise, poor time management, and escalating academic stress also contribute to weight gain [48]. Besides, people who are overweight or obese are more likely to frequent fast food joints and cafes, which may indicate that they eat more foods that are heavy in sugar, salt, and fat. The convenience of fast-food restaurants increases the likelihood that students will overeat sugary drinks, fatty foods, and high-energy foods while decreasing their consumption of healthy options. When time is of the essence and there is a heavy course load, fast food is an inexpensive and convenient option for university students [49]. These studies clarify the observed positive correlation between BMI score and SHS among university students in our study.

In addition to the above-identified unhealthy lifestyle as SHS predictors, age stood out as another predictor besides the college and academic level. We found that an increase in age resulted in a decrease in SHS score, indicating that younger age had a higher risk of SHS. The findings of this study are incongruous with previous research, which indicated that younger adolescents exhibited superior health conditions [50]. The primary factor contributing to this phenomenon is that a significant proportion of young students have experienced immense pressure to achieve high scores in the early years of their university studies. Furthermore, many students engage in prolonged periods of sedentary studying, seldom venturing outdoors for physical activity and occasionally neglecting their dietary and nutritional needs, thereby compromising their overall health. Captivatingly, our study reveals a positive correlation between academic level and SHS, indicating that students at higher levels of their studies are at higher risk of SHS. This could be partly attributed to poor eating habits which progress with advance in academic level. Nelson et al. (1991) had earlier observed weight gain throughout a university's first years and significant overweight and obesity in the subsequent years [50]. Moreover, several studies reported poor health habits among university students, usually initiated in their early years of university studies [51].

Our study has significant implications and validates the need for policies and program towards achieving a healthy lifestyle for university students and other populations. The government of Saudi Arabia has already set the goal "to increase the ratio of individuals exercising at least once a week from 13% of population to 40%" by 2030. These goals can be achieved if the physical activity awareness campaigns are interspersed with recreational activities. Moreover, the Saudi society is becoming more receptive to adopt global trends encourage the exchange of ideas, knowledge, and experiences to promote wellbeing and quality of life for

Saudi population. This transitional phase is a promising opportunity to inculcate healthy habits in youth and overcome the current healthcare burden and challenges due to high prevalence of chronic disease in Saudi Arabia.

Our study had some limitations that should be noted to interpret results. Firstly, the cross-section design makes it impossible to determine causality due to its single-point data collection. Secondly, SHS lacks objective clinical diagnosis and is subjective. Thirdly, data collection via an online self-report questionnaire may be influenced by self-reporting. However, the large sample size recruited in this present study has contributed to the consistency of the findings. This can be replicated in other regions and countries to add valuable insights and contribute to academia.

Future research could explore the effectiveness of interventions aimed at promoting healthy lifestyles and reducing SHS among university students. Qualitative research could provide deeper insights into the reasons behind students' sedentary behaviors and identify potential barriers to adopting healthy lifestyles.

## 5. Conclusion

In this study, the key predictors contributing to SHS among undergraduate students in Saudi Arabia are brought to the forefront. Notably, modifiable predictors such as poor sleep quality, higher BMI scores, and poor dietary habits increase SHS risk among university students in Saudi Arabia. Therefore, the findings of this study emphasize the necessity of early interventions that promote healthy lifestyles to enhance the academic performance and well-being of university students, thereby preventing or delaying the onset of chronic diseases. Moreover, more research is needed to explore the effectiveness of interventions aimed at promoting healthy lifestyles and reducing SHS among university students.

## Supporting information

**S1 Data. Undergraduates lifestyle and SHS 2023.**
(SAV)

## Author Contributions

**Conceptualization:** Bandar Alsaif, Collins Otieno Asweto, Sehar-un-Nisa Hassan, Mohamed Ali Alzain.

**Data curation:** Collins Otieno Asweto, Mohammed Elshiekh Saeed, Kamal Elbssir Mohammed Ali, Rafat Zrieq.

**Formal analysis:** Collins Otieno Asweto, Mohamed Ali Alzain.

**Funding acquisition:** Bandar Alsaif, Collins Otieno Asweto, Sehar-un-Nisa Hassan, Mohamed Ali Alzain, Mohammed Elshiekh Saeed, Ahmed Kassar, Kamal Elbssir Mohammed Ali.

**Investigation:** Bandar Alsaif, Collins Otieno Asweto, Sehar-un-Nisa Hassan, Mohamed Ali Alzain, Mohammed Elshiekh Saeed, Ahmed Kassar.

**Methodology:** Bandar Alsaif, Collins Otieno Asweto, Sehar-un-Nisa Hassan, Mohamed Ali Alzain, Wei Wang.

**Project administration:** Bandar Alsaif, Mohammed Elshiekh Saeed, Ahmed Kassar, Kamal Elbssir Mohammed Ali, Mouna Ghorbel, Rafat Zrieq.

**Resources:** Mohammed Elshiekh Saeed, Ahmed Kassar, Kamal Elbssir Mohammed Ali, Mouna Ghorbel, Rafat Zrieq, Wei Wang.

**Software:** Collins Otieno Asweto, Mouna Ghorbel.

**Supervision:** Bandar Alsaif, Collins Otieno Asweto, Sehar-un-Nisa Hassan, Mohamed Ali Alzain, Wei Wang.

**Validation:** Bandar Alsaif, Sehar-un-Nisa Hassan, Mohamed Ali Alzain, Mohammed Elshiekh Saeed, Kamal Elbssir Mohammed Ali, Rafat Zrieq.

**Visualization:** Bandar Alsaif, Collins Otieno Asweto, Mohamed Ali Alzain.

**Writing – original draft:** Collins Otieno Asweto, Sehar-un-Nisa Hassan, Mohamed Ali Alzain.

**Writing – review & editing:** Bandar Alsaif, Collins Otieno Asweto, Sehar-un-Nisa Hassan, Mohamed Ali Alzain, Ahmed Kassar, Kamal Elbssir Mohammed Ali, Mouna Ghorbel, Rafat Zrieq, Wei Wang.

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
