## [Decision Letter · Decision Letter 0]

7 Oct 2024

PONE-D-24-20915Undergraduates’ lifestyle and suboptimal health status (SHS): A cross-sectional study in the Ha’il region of Saudi ArabiaPLOS ONE

Dear Dr. Hassan,

Thank you for submitting your manuscript to PLOS ONE. After careful consideration, we feel that it has merit but does not fully meet PLOS ONE’s publication criteria as it currently stands. Therefore, we invite you to submit a revised version of the manuscript that addresses the points raised during the review process.

We look forward to receiving your revised manuscript.

Kind regards,

Mukhtiar Baig, Ph.D.

Academic Editor

PLOS ONE

Journal Requirements: When submitting your revision, we need you to address these additional requirements. 1. Please ensure that your manuscript meets PLOS ONE's style requirements, including those for file naming. The PLOS ONE style templates can be found at https://journals.plos.org/plosone/s/file?id=wjVg/PLOSOne_formatting_sample_main_body.pdf and https://journals.plos.org/plosone/s/file?id=ba62/PLOSOne_formatting_sample_title_authors_affiliations.pdf 2. Thank you for stating the following financial disclosure: "This research was funded by the Deanship of Scientific Research at the University of Ha’il, funding number RG-20 215." Please state what role the funders took in the study.  If the funders had no role, please state: ""The funders had no role in study design, data collection and analysis, decision to publish, or preparation of the manuscript."" If this statement is not correct you must amend it as needed. Please include this amended Role of Funder statement in your cover letter; we will change the online submission form on your behalf. 3. In this instance it seems there may be acceptable restrictions in place that prevent the public sharing of your minimal data. However, in line with our goal of ensuring long-term data availability to all interested researchers, PLOS’ Data Policy states that authors cannot be the sole named individuals responsible for ensuring data access (http://journals.plos.org/plosone/s/data-availability#loc-acceptable-data-sharing-methods). Data requests to a non-author institutional point of contact, such as a data access or ethics committee, helps guarantee long term stability and availability of data. Providing interested researchers with a durable point of contact ensures data will be accessible even if an author changes email addresses, institutions, or becomes unavailable to answer requests. Before we proceed with your manuscript, please also provide non-author contact information (phone/email/hyperlink) for a data access committee, ethics committee, or other institutional body to which data requests may be sent. If no institutional body is available to respond to requests for your minimal data, please consider if there any institutional representatives who did not collaborate in the study, and are not listed as authors on the manuscript, who would be able to hold the data and respond to external requests for data access? If so, please provide their contact information (i.e., email address). Please also provide details on how you will ensure persistent or long-term data storage and availability. 4. Please include captions for your Supporting Information files at the end of your manuscript, and update any in-text citations to match accordingly. Please see our Supporting Information guidelines for more information: http://journals.plos.org/plosone/s/supporting-information.

Reviewers' comments:

Reviewer's Responses to Questions

**Comments to the Author**

1. Is the manuscript technically sound, and do the data support the conclusions?

Reviewer #1: Yes

Reviewer #2: Yes

Reviewer #3: No

2. Has the statistical analysis been performed appropriately and rigorously? 

Reviewer #1: Yes

Reviewer #2: Yes

Reviewer #3: Yes

3. Have the authors made all data underlying the findings in their manuscript fully available?

Reviewer #1: Yes

Reviewer #2: No

Reviewer #3: No

4. Is the manuscript presented in an intelligible fashion and written in standard English?

Reviewer #1: Yes

Reviewer #2: Yes

Reviewer #3: No

5. Review Comments to the Author

Reviewer #1: Please write nomenclature of SHS in the abstract.Conclusion needs to be rephrased in accordance with the objectives.The researchers employed a well-designed cross-sectional study to collect data from a large sample of undergraduate students and used appropriate statistical methods to analyze the findings.

Key strengths of the study:

Large sample size: The study included a substantial number of participants, which increases the generalizability of the findings.

Comprehensive data collection: The researchers used validated questionnaires to assess SHS, lifestyle factors, and sociodemographic characteristics.

Rigorous statistical analysis: The study employed appropriate statistical methods to analyze the data and identify significant associations.

Clear presentation of results: The results are clearly presented and supported by statistical evidence.

Areas for improvement:

Intervention study: Future research could explore the effectiveness of interventions aimed at promoting healthy lifestyles and reducing SHS among university students.

Qualitative research: Qualitative research could provide deeper insights into the reasons behind students' sedentary behaviors and identify potential barriers to adopting healthy lifestyles.

Overall, the study makes a valuable contribution to the literature on health behaviors among young people and provides important implications for policymakers and health professionals in Saudi Arabia. The findings highlight the need for targeted interventions to promote healthy lifestyles and prevent the negative consequences of SHS among university students.

Reviewer #2: Its a very well written manuscript. I would like to congratulate the team for a great work on a very essential area. This can be replicated in other regions and countries to add valuable insights and contribute to the academia.

Reviewer #3: This paper aims to shed light on the predictive factors that impact SHS experienced by students studying in higher education in Saudi Arabia. The authors use two instruments: ASHSQ-25 and SLIQ to capture data from 9,026 individuals. I agree that this is an important topic, but I have substantial concerns about its theoretical and practical contributions.

1. The manuscript does not offer a theoretical contribution to how we think about population health. It also doesn’t offer substantive insights (the main contribution is the implementation of SHSQ-25 in Saudi Arabia). The data presented is correlational, meaning it cannot be used to draw causal inferences.

2. Data - I understand the restrictive nature of the data used in this investigation, the ongoing crisis the scientific community has been dealing with in recent years necessitates transparency, encompassing all aspects related to methods, data, and analyses. Having reviewed enough manuscripts, I know it is possible to uphold transparency while protecting participants, companies, and data.

3. Health, both emotional and physical, is important across income and education levels and ages. As such, I find the authors’ motivation for this investigation–focusing on college students–unfortunate. Even if the data is limited to a certain age group, the authors could probably consider a broader target population. If I’m wrong, and there is something special about college students, the authors should offer compelling arguments.

4. Finally, describing SHSQ-25 as a “…well-known self-report measure that has been well-tested for its psychometric properties and has been widely used in the assessment of SHS” is somewhat inaccurate. A search in Google Scholar produced only 209 citations for the original paper (2009), some of which were self-citations.

6. PLOS authors have the option to publish the peer review history of their article (what does this mean?). If published, this will include your full peer review and any attached files.

Reviewer #1: **Yes: **Rehana Rehman

Reviewer #2: **Yes: **Raafat Hassan

Reviewer #3: No

---

## [Author Response · Author response to Decision Letter 0]

12 Dec 2024

Subject: Response to the Editor and Reviewers’ Comments on research article submitted for consideration for publication at PLOS ONE. 

Original Research Article 

PONE-D-24-20915

Undergraduates’ lifestyle and suboptimal health status (SHS): A cross-sectional study in the Ha’il region of Saudi Arabia

We thank you for the review and valuable comments on our manuscript. Please find below the point-by-point response to both editorial and reviewers' comments. We hope for your kind consideration. 

Editorial Requirement 1: A rebuttal letter that responds to each point raised by the academic editor and reviewer(s). You should upload this letter as a separate file labelled 'Response to Reviewers'.

Response 1: Done. 

Editorial Requirement 2: A marked-up copy of your manuscript highlighting changes made to the original version. You should upload this as a separate file labelled 'Revised Manuscript with Track Changes'.

Response 2: The manuscript with changes highlighted in yellow has been submitted as a separate file. 

Editorial Requirement 3: An unmarked version of your revised paper without tracked changes. You should upload this as a separate file labelled 'Manuscript'.

Response 3 : An unmarked copy of the manuscript has been uploaded as well. 

Editorial Requirements 4: Please ensure that your manuscript meets PLOS ONE's style requirements, including those for file naming. 

Response 4:The manuscript has been transformed according to the template requirements as per guidelines provided in the templates. 

Editorial Requirements 5: Please state what role the funders took in the study. If the funders had no role, please state: "The funders had no role in study design, data collection and analysis, decision to publish, or preparation of the manuscript."" 

Response 5: Yes, the funder has no role, and the required adjustment has been made in the funding statement. 

Editorial Requirement 6: Data requests to a non-author institutional point of contact, such as a data access or ethics committee, help guarantee long term stability and availability of data. Providing interested researchers with a durable point of contact ensures data will be accessible even if an author changes email addresses, institutions, or becomes unavailable to answer requests.

Response 6: Thank you for this important point. We agree that appropriate measures must be taken to guarantee long term stability and availability of data. We have provided a non-author institutional point of contact as well in the data availability statement. 

“Access to anonymous data used for this analysis will be available upon direct request and sent to either the corresponding author on this manuscript and/or the secretary of training at the College of Public Health and Health Informatics via e-mail at: training.cphhi@uoh.edu.sa . 

Reviewers Comments

Response to Reviewer 1 Comments

Comment #1: Please write the nomenclature of SHS in the abstract. The conclusion needs to be rephrased in accordance with the objectives.

Response 1: Adjusted the changes. (Pg 3; Line 35). Rephrased the conclusion (Pg 3 Line 46-48).

Comment # 2: Add future research focus on intervention study: Future research could explore the effectiveness of interventions aimed at promoting healthy lifestyles and reducing SHS among university students. Qualitative research: Qualitative research could provide deeper insights into the reasons behind students' sedentary behaviours and identify potential barriers to adopting healthy lifestyles.

Response 2: Added the directions for future research as per the above recommendation by the reviewer 1 (Pg 15 Line 319-321).

Response to Comments by Reviewer 2 

Comment #1: It's a very well-written manuscript. I would like to congratulate the team for their great work on a very essential area. This can be replicated in other regions and countries to add valuable insights and contribute to academia.

Response #1: Thank you for your appreciation. We have added the point recommended by you in limitations and directions for future research. (Pg 15 Line 317-321).

Response to Comments by Reviewer 3

Comment 1 (Part 1): The manuscript does not offer a theoretical contribution to how we think about population health. It also doesn’t offer substantive insights (the main contribution is the implementation of SHSQ-25 in Saudi Arabia). The data presented is correlational, meaning it cannot be used to draw causal inferences. 

Response (Part 1): Thank you for this important comment and appreciate it. We made some modifications in the introduction and have added a section with some relevant points on the theoretical framework on population health and SHS (Pg 4; Line 67-81)

Response (Part 2): Appreciate your observation and agree that the study is correlational and drawing causal inferences is not asserted in this manuscript. The title also refers to the current research as a cross-sectional study, however, the regression models can be used to do the predictive analysis based on this cross-sectional data. 

Comment 2: Data - I understand the restrictive nature of the data used in this investigation, the ongoing crisis the scientific community has been dealing with in recent years necessitates transparency, encompassing all aspects related to methods, data, and analyses. Having reviewed enough manuscripts, I know it is possible to uphold transparency while protecting participants, companies, and data.

Response 2: Thank you for this important comment. We have added in the data availability statement the possibility to access anonymous data directly from the authors or through the institution (Pg 16; Line 333-335). 

Data Availability Statement: Access to anonymous data used for this analysis will be available upon direct request sent via e-mail to either the corresponding author on this manuscript and/or the secretary of training at the College of Public Health and Health Informatics at: training.cphhi@uoh.edu.sa

Comment 3: Health, both emotional and physical, is important across income and education levels and ages. As such, I find the authors’ motivation for this investigation–focusing on college students–unfortunate. Even if the data is limited to a certain age group, the authors could probably consider a broader target population. If I’m wrong, and there is something special about college students, the authors should offer compelling arguments. 

Response 3. Thank you very much for this important comment. We agree that income, education and age are important determinants of health. This project was carried out as part of the bigger project that has collected data from target populations including the general population with a wide range of age, education and income levels. However, in this analysis, we particularly focused on university students for the following reasons which we have mentioned and further elaborated in the manuscript (Pg 3; Line 51-54); (Pg 5-6; Line 108-119). 

Comment 4: Finally, describing SHSQ-25 as a “…well-known self-report measure that has been well-tested for its psychometric properties and has been widely used in the assessment of SHS” is somewhat inaccurate. A search in Google Scholar produced only 209 citations for the original paper (2009), some of which were self-citations.

Response 4: Thank you for this important point. We have rephrased this part of the manuscript in light of your comments. (Pg 7 ; Line 147-149).

---

## [Editor Report · Decision Letter 1]

22 Dec 2024

Undergraduates’ lifestyle and suboptimal health status (SHS): A cross-sectional study in the Ha’il region of Saudi Arabia

PONE-D-24-20915R1

Dear Dr. Hassan,

We’re pleased to inform you that your manuscript has been judged scientifically suitable for publication and will be formally accepted for publication once it meets all outstanding technical requirements.

Kind regards,

Mukhtiar Baig, Ph.D.

Academic Editor

PLOS ONE
---

## [Editor Report · Acceptance letter]

9 Jan 2025

PONE-D-24-20915R1 

PLOS ONE

Dear Dr. Hassan, 

I'm pleased to inform you that your manuscript has been deemed suitable for publication in PLOS ONE. Congratulations! Your manuscript is now being handed over to our production team.

Kind regards, 

on behalf of

Professor Mukhtiar Baig 

Academic Editor

PLOS ONE